# EnNet: Enhanced Interactive Information Network with Zero-Order Optimization

**DOI:** 10.3390/s24196361

**Published:** 2024-09-30

**Authors:** Yingzhao Shao, Yanxin Chen, Pengfei Yang, Fei Cheng

**Affiliations:** 1State Key Laboratory of Integrated Services Networks, Xidian University, Xi’an 710071, China; shaoyz@cast504.com; 2Key Laboratory of Smart Human-Computer Interaction and Wearable Technology of Shaanxi Province, Xi’an 710071, China; 3School of Computer Science and Technology, Xidian University, Xi’an 710126, China; chengfei@xidian.edu.cn

**Keywords:** interactive image segmentation, self-attention mechanism, global features, fine-grained features, semi-supervised optimization

## Abstract

Interactive image segmentation extremely accelerates the generation of high-quality annotation image datasets, which are the pillars of the applications of deep learning. However, these methods suffer from the insignificance of interaction information and excessively high optimization costs, resulting in unexpected segmentation outcomes and increased computational burden. To address these issues, this paper focuses on interactive information mining from the network architecture and optimization procedure. In terms of network architecture, the issue mentioned above arises from two perspectives: the less representative feature of interactive regions in each layer and the interactive information weakened by the network hierarchy structure. Therefore, the paper proposes a network called EnNet. The network addresses the two aforementioned issues by employing attention mechanisms to integrate user interaction information across the entire image and incorporating interaction information twice in a design that progresses from coarse to fine. In terms of optimization, this paper proposes a method of using zero-order optimization during the first four iterations of training. This approach can reduce computational overhead with only a minimal reduction in accuracy. The experimental results on GrabCut, Berkeley, DAVIS, and SBD datasets validate the effectiveness of the proposed method, with our approach achieving an average NOC@90 that surpasses RITM by 0.35.

## 1. Introduction

Image datasets with high-quality segmentation annotations play a crucial role in computer vision applications such as autopilot [1], remote sensing [2], etc. These segmentation annotations also have the potential to be utilized by fundamental downstream tasks, including object detection [3] and object tracking [4]. To quickly obtain high-quality annotations, interactive image segmentation is designed to extract regions of interest (ROI) from an image using user annotation information as prior knowledge, reducing the expensive cost of manual labor. Specifically, the popular paradigm of this technique simulates the action of human interactive (e.g., click, stroke, and bounding box) as the response maps and trains the networks with response maps, image data, and their mask labels by supervised learning. For new images, we can simply click on ROI, and the pixel-level annotations could be generated by an interactive segmentation network iteratively.

However, existing methods suffer from insufficient utilization of interaction information and excessively high computational burden in optimization. The inadequate utilization of interactive information stems from insufficient information mining and information decay, leading to issues of over-segmentation and boundary blur, respectively. The high computational burden in optimization is attributed to the fact that existing optimization procedures are nearly equivalent to retraining, resulting in an exorbitant optimization cost. The motivation behind this work is to address these limitations by proposing a method that improves the utilization of interactive information and reduces the computational burden of optimization, making interactive image segmentation more efficient and effective for real-world applications. The following is a detailed analysis from three aspects:

Over-segmentation: Over-segmentation occurs when a large part of the background region is mistakenly segmented as the foreground region in Figure 1a. This issue arises from the similar features extracted by the learning algorithm from these two regions regardless of the actual circumstances. Specifically, homogeneous regions [5] and insufficient feature interpretation [6] are the two primary sources of similar features. Homogeneous regions mean that the background resembles the texture of the target region, while insufficient features mean that the segmentation algorithms are unable to establish the distinct features of background and foreground regions. Fortunately, interactive information in interactive segmentation tasks clearly indicates with response maps whether the pixels of small click regions are the background or foreground. The exploration of semantic information will be conducted through the utilization of response maps and pixel values for the reduction of over-segmentation. If this exploration does not go well, the segmentation result may not align with human expectations. According to Lin et al. [7], existing methods treat all user interaction information equally, disregarding the distinction between initial interaction information and subsequent interactions. This oversight leads to an increased number of necessary interaction cycles. To address this, they propose FCA-Net, which aims to minimize the need for subsequent clicks. However, the fundamental reason for this issue lies in the fact that existing interactive image segmentation methods mostly rely on fully convolutional neural networks (FCNs) for feature extraction. The local nature of convolutional operations limits the network’s receptive field [8,9], resulting in a lack of global information in extracted features. Specifically in interactive image segmentation, this results in less representative features for each interactive region, leading to large-scale segmentation errors.

Boundary blur: Boundary blur denotes that the segmented boundary between foreground and background is jagged, as shown in Figure 1b, not following the smooth style. This problem stems from the loss of the interaction information increasing with the network hierarchy structure. Although the relationship of interactive and pixel value as the output feature map has been established in each convolution layer, the process of down-sampling in each layer of the network diminishes the interactive information to a certain degree. The severity of this issue worsens with each subsequent layer. Several segmentation techniques attempt to resolve the problem of blurry segmentation boundaries. Cheng et al. [10] proposed BMask R-CNN, employing the creation of multiple branches to counteract information loss on the main branch. Marin et al. [11] introduced a new content-adaptive downsampling technique to reduce the loss of fine-grained details during downsampling. In interactive image segmentation, the weakening of user interaction information and the image along the network hierarchy can result in the inability to accurately delineate the foreground and background boundaries in the segmentation region, even with a substantial amount of added interaction information.

Costly optimization: Existing methods [12,13] involve fine-tuning a subset of parameters in the model after completing the training process to enhance the segmentation efficiency of challenging target regions. However, this optimization approach incurs a cost comparable to retraining the model from scratch. Although the trained network can provide accurate segmentation results with just a few clicks for images containing known objects (such as humans or motorcycles) present in the training set, it fails to deliver satisfactory results for unknown objects(such as humans and motorcycles), even with a large amount of user input, as shown in Figure 1c. For such challenging objects that are difficult to segment, this paper adopts a semi-supervised optimization method similar to f-BRS [13] to address this issue. While f-BRS offers improvements in terms of speed compared with BRS [12], it still suffers from high time costs. To mitigate this, the paper introduces zero-order optimization based on the foundation of semi-supervised optimization methods to reduce optimization costs while maintaining segmentation accuracy.

For the above three issues, the main contributions of this paper are as follows:A feature augmentation module based on a self-attention mechanism is introduced to enhance the guiding role of interactive information across the entire image.Combining the boundary optimization module with PointRend enables the network to fully compensate for lost information and extract detailed local information from the input. This addresses the issue of information degradation caused by the network’s hierarchical structure.A semi-supervised prediction optimization method has been proposed to reduce time costs by leveraging zero-order optimization. This method utilizes interactive clicks as supervisory information to adjust the output values in the model, thereby enhancing the guiding role of interactive information in segmentation and improving the accuracy and efficiency of segmenting challenging target regions.

Due to the high cost of collecting interactive information, the interaction data used during model training is simulated by a computer. However, in real-world applications, user interaction information needs to be obtained through sensors such as mice, touchscreens, and eye trackers. The user interaction data collected through these sensors enriches the model’s understanding of user intent and enhances the results of image processing. Additionally, this sensor-acquired interaction data can be used for subsequent model fine-tuning, leading to improved segmentation performance.

The rest of the article is organized as follows: In Section 2, we will review related work and introduce some interactive image segmentation methods from the past and present. In Section 3, we will present the proposed method. In Section 4, we will show a series of experimental results. In Section 5, we will conclude the paper.

## 2. Related Works

In the related work, research on interactive image segmentation can be broadly categorized into two directions: classical methods and deep learning methods. Classical methods typically rely on traditional computer vision techniques and image processing algorithms, requiring extensive manual interaction to achieve image segmentation. On the other hand, deep learning methods leverage the powerful representation and learning capabilities of deep neural networks, training on large annotated datasets to enhance segmentation efficiency. This section provides a brief introduction and comparison of research in these two directions.

### 2.1. Classical Methods

Before the deep learning-based interactive segmentation was proposed, interactive segmentation was defined as an optimization problem. Optimization-based interactive image segmentation methods include graph cut-based methods [14,15], contour-based methods [16], random walk-based methods [17,18] and region-merging-based methods [19,20]. Graph cuts [14] transform the image segmentation problem into a minimum cut problem and divide the image into two or more regions by calculating the minimum cut in the graph. Intelligent scissors [21] is one of the earliest contour-based methods, which uses the Dijkstra algorithm to compute the shortest path connecting the seed points to extract the target contour. The random walk algorithm [17,18] models the image as a graph, where each pixel is a node in the graph and adjacent pixels are connected by edges. Then, by giving labels to some seed pixels and using the random walk algorithm to iteratively expand the region of pixels with the same label on the graph, image segmentation is achieved. The region-growing algorithm [19,20] first selects one or more seed points, computes the similarity between the seed points and adjacent pixels, and then adds pixels with similarity exceeding a certain threshold to the same region as the seed points. The algorithm gradually expands the range of the region until all pixels are assigned to one region or no more pixels can be added to any region. Due to the lack of guided high-level semantic information, optimization-based interactive image segmentation methods often require a significant amount of interaction information for assistance, resulting in low efficiency.

### 2.2. Deep Learning Methods

Since Xu et al. [22] first utilized deep learning in interactive segmentation, significant improvements have been achieved. Deep learning-based interactive segmentation methods [7,12,13,22,23,24,25,26,27,28,29,30,31,32,33,34,35,36] have gradually become mainstream. There exists a group of researchers who have devised diverse network architectures aimed at enhancing the model’s comprehension of image semantics and user interaction information in order to elevate the precision and efficiency of segmentation. On the other hand, a separate group of scholars has proposed universal optimization methods applicable to various network architectures, enabling the model to achieve superior segmentation outcomes while keeping the network structure unchanged.

Network architecture: Li et al. [23] proposed the Latent Diversity algorithm, which generates multiple segmentation results using a convolutional neural network after receiving interaction clicks and selects the optimal result via a selection network. Liew et al. [26] introduced RIS-Net, a dual-branch network that uses a global branch for initial segmentation and a local branch for refining regions near interactive clicks. The results from both branches are fused and optimized using graph cuts for final segmentation. Unlike these approaches, our method integrates a global feature enhancement module to better capture global contextual information, improving the effectiveness of interactive inputs across the entire image. EdgeFlow [28] highlighted the risk of diluting interaction information by fusing it too early with image features, proposing an early-late fusion strategy with repeated interaction feature integration for better accuracy. However, as the network deepens, early-stage interaction information can still degrade, leading to poor boundary segmentation. To address this, we propose FineNet, which reintegrates information in later stages to maintain strong guidance from interaction inputs throughout the network.

Optimization procedure: Existing optimization methods either simulate real user workflows during training or fine-tune models using user-annotated data after training to improve accuracy or efficiency without changing the network structure. Mahadevan et al. [25] introduced Iteratively Trained Interactive Segmentation (ITIS), which uses simulated clicks during training based on segmentation errors, closely mimicking user interactions. Jang et al. [12] proposed BRS, which corrects mislabeled pixels during inference based on user clicks but requires multiple forward and backward passes, making it computationally expensive. To reduce this cost, Sofiiuk et al. [13] developed f-BRS, which adjusts the channel weights of feature maps using backpropagation and forward propagation. While f-BRS is more efficient, even small changes in channel weights can cause large-scale segmentation errors, leading to inefficiencies. Our proposed method addresses this by modifying only a small set of model parameters, improving segmentation efficiency for difficult targets without compromising accuracy. Additionally, zero-order optimization is used to further speed up the process.

## 3. Methodolgy

In this paper, our work contains network architecture and optimization procedures. Firstly, we propose a multi-scale feature-based interactive image segmentation network named EnNet, which adopts a coarse-to-fine network structure, as shown in Figure 2. The EnNet consists of CoarseNet and FineNet. The input image and interactive clicks are fed into CoarseNet to generate coarse logits, and then FineNet is used to refine the mask. Secondly, in the inference stage, a semi-supervised prediction optimization algorithm is applied to modify the intermediate output values of EnNet, correcting the misclassified pixels in the region near the interactive points.

### 3.1. Input

Interactive image segmentation, unlike other segmentation tasks, involves user guidance. Therefore, it requires encoding user annotation information and integrating it with the image before inputting it into the network. The following are the click encoding and fusion modules:

Clicks encoding: The interaction information includes positive and negative clicks, where positive clicks indicate the target region and negative clicks indicate the background. Sofiiuk et al. [37] conducted an ablation study to compare distance transform encoding with disk encoding and found that disk encoding was superior. Therefore, the clicks are encoded as an interaction click map, with a circle point of radius five pixels used to encode positive and negative clicks, generating a two-channel click map with the same size as the image.

Fusion module: The input fusion module fuses the image and interaction information to generate the input information for the network. The segmentation result generated from the previous interaction is used as additional interaction information and is concatenated along the channel dimension with the click map to generate a three-channel interaction map. Finally, the image and interaction map are fused by increasing their dimensions with 1×1 and 3×3 convolutions and added pointwise. The process of input fusion can be seen in Figure 3.

### 3.2. CoarseNet

CoarseNet consists of a basic segmentation network and a global feature enhancement module. We built a basic segmentation network based on DeepLabV3+ [38], which consists of an input fusion module, a backbone network ResNet101 [39], an ASPP module, and a decoder.

Fully convolutional neural networks have fixed receptive fields and translation invariance, which may make it challenging for them to capture globally effective contextual information and exhibit poor sensitivity to specific locations when compared with attention mechanisms. Therefore, in interactive image segmentation using fully convolutional neural networks, it may be challenging to achieve accurate understanding of image semantics and user interaction information, leading to significant errors in segmenting target regions in practical applications. In this paper, we introduce a Global Feature Enhancement Module (GFEM) based on self-attention mechanism to enhance the global contextual information aggregation capability of the model, enrich the representative features of interactive regions in each layer, and make the model more accurate in selecting target regions. The GFEM module consists of an embedding layer and three layers of Transformer encoders, as shown in Figure 4.

The working process of the GFEM module is as follows. The feature map $F_{in}$ with a size of H×W×2048 from the backbone network is fed into the embedding layer, followed by a 1×1 convolutional operation to obtain a feature map $F_{0}$ with a size of H×W×256. Since the encoder requires sequential data as input, the spatial dimensions of $F_{0}$ are folded into one dimension to obtain a feature sequence $z_{0}$ with a dimension of H×W×256. Positional encoding is added to $z_{0}$ to preserve spatial position information, as shown in Equation (Equation 1).
(1)$$\begin{array}{cc} z_{0} & =Reshape\left(Conv_{1\times 1}\left(F_{in}\right)\right)+E_{pos} \end{array}$$

Then, $z_{0}$ is fed into the encoder to calculate multi-head self-attention and obtain $z_{l}^{\prime }$, as shown in Equation (Equation 2).
(2)$$\begin{array}{cccc} z_{l}^{\prime } & =\mathrm{MSA}\left(\mathrm{LN}\left(z_{l-1}\right)\right)+z_{l-1}, & \; & l=1\ldots L \end{array}$$

After layer normalization, $z_{l}$ is obtained through MLP, as shown in Equation (Equation 3).
(3)$$\begin{array}{cccc} z_{l} & =\mathrm{MLP}\left(\mathrm{LN}\left(z_{l}^{\prime }\right)\right)+z_{l}^{\prime }, & \; & l=1\ldots L \end{array}$$

Repeat this step *L* times to obtain the output of the last layer $z_{L}$ of the encoder. Finally, after layer normalization, $z_{out}$ is obtained and reshaped into the feature map $F_{out}$ with a size of H×W×256, as shown in Equation (Equation 4).
(4)$$\begin{array}{ccc} F_{out} & =Reshape\left(\mathrm{LN}\left(z_{L}^{0}\right)\right) & \; \end{array}$$

In the above equation, $Conv_{1\times 1()}$ represents the 1×1 convolution, $E_{pos}$ represents the position encoding, MSA() represents the multi-head self-attention, MLP() represents the MLP layer, LN() represents the layer normalization, and Reshape() represents the reshaping operation.

### 3.3. FineNet

The multiple downsampling operations in CoarseNet lead to a weakening of the guidance of interactive information and the loss of image information. Consequently, the segmentation results produced by CoarseNet exhibit inaccuracy at the boundaries of the target region. In this paper, we propose FineNet to solve this problem. FineNet consists of an edge refinement module (ERM) based on fine-grained features. ERM further refines the boundaries of the coarse segmentation result to obtain a more accurate fine mask. As shown in Figure 5, ERM consists of a convolution-based fine-grained feature extraction module and PointRend [40]. This module concatenates the image, click map, and coarse logits along the channel dimension as input, first uses a shallow convolutional network to extract fine-grained features, and then uses PointRend to optimize the boundaries of the coarse logits using rich detail information in the fine-grained features, thereby obtaining a more accurate boundary segmentation result.

The detailed processing steps of ERM are as follows. The input of ERM includes the image, positive and negative interactive clicks, and coarse logits. This helps to restore the lost information that occurs during extensive downsampling. As larger-sized disks are not suitable for indicating the boundaries of the target region, disks with a radius of two pixels are used to encode positive and negative clicks, generating a two-channel click map with the same size as the image. The original image, click map, and coarse logits are concatenated along the channel dimension to generate a six-channel input information map. The process of input information processing is shown in Equation (Equation 5), where *I* represents the original image, $M_{c}$ represents the click map, and $M_{l}$ represents the coarse segmentation result.
(5)$$\begin{array}{cc} X & =Concat(I,M_{c},M_{l}) \end{array}$$

As shallow networks have smaller receptive fields and smaller overlapping regions, they ensure that the network captures more details. ERM module uses a shallow convolutional network to extract fine-grained features, as shown in Equation (Equation 6). The input information *X* is extracted by five convolutional blocks to obtain fine-grained features $F_{l}$.
(6)$$\begin{array}{cc} F_{l}=\; & Conv_{1\times 1}(Conv_{3\times 3}(Conv_{1\times 1}(\\ & Conv_{3\times 3}(Conv_{3\times 3}(X))))) \end{array}$$

PointRend is an upsampling method for image segmentation, designed to enhance the segmentation performance of object boundaries by leveraging detailed features.

### 3.4. Loss Function

The EnNet consists of CoarseNet and FineNet. CoarseNet focuses on the overall target region, while FineNet is more focused on the boundaries of the target region. Considering the differences mentioned above, different parts should use different loss.

CoarseNet uses the Normalized Focal Loss (NFL) as shown in Equation (Equation 7).
(7)$$\begin{array}{cc} l_{N}\left(i,j\right) & =-\frac{1}{\Sigma_{i,j}{\left(1-pt_{i,j}\right)}^{\gamma }}{\left(1-pt_{i,j}\right)}^{\gamma }\log(pt_{i,j}) \end{array}$$
γ is an adjustable factor greater than 0, and pt(i,j) represents the predicted confidence of the pixel point (i,j), defined as shown in Equation (Equation 8). p(i,j) is the predicted value of pixel point (i,j), and *y* is the ground truth label of pixel point (i,j), with y=1 indicating the target region and y=0 indicating the background. pt(i,j) reflects the difficulty level of pixel classification, with a higher pt(i,j) indicating a higher confidence and easier classification, while a lower pt(i,j) indicating a lower confidence and more difficult classification. Therefore, NFL increases the weight of hard-to-segment samples in the loss, which helps to improve the accuracy of hard samples.
(8)$$\begin{array}{c} pt_{i,j}=\left\{\begin{array}{cc} p, & y=1\\ 1-p, & y=0 \end{array}\right. \end{array}$$

FineNet is designed to improve the segmentation performance of the model for the object boundaries. Therefore, the loss function of FineNet needs to focus more on the pixels on the object boundaries. To this end, ERM adopts a normalized focal loss with boundary weighting in the calculation of the loss. The pixels on the object boundaries are assigned higher weights, as shown in Equation (Equation 9). Specifically, $\left(i,j\right)\in O_{e}$ indicates that the pixel (i,j) is on the object boundaries, while $\left(i,j\right)\in O_{b}$ indicates that it is another pixel. The weight of the loss caused by the pixels on the object boundaries, denoted by α, is set to 1.5.
(9)$$\begin{array}{c} l_{W}\left(i,j\right)=\alpha \underset{\left(i,j\right)\in O_{e}}{L_{N}\left(i,j\right)}+\underset{\left(i,j\right)\in O_{b}}{L_{N}\left(i,j\right)} \end{array}$$

The Point Head in the PointRend module of ERM is supervised by binary cross entropy, as shown in Equation (Equation 10), where $P_{e}$ represents the set of points selected by PointRend for optimization.
(10)$$\begin{array}{cc} l_{B}= & -\frac{1}{N}\sum_{\left(i,j\right)\in P_{e}}[y\left(i,j\right)\cdot log(p(i,j))\\ & +(1-y(i,j))\cdot log(1-p(i,j))] \end{array}$$

The final loss function is formulated as Equation (Equation 11).
(11)$$\begin{array}{c} L=l_{N}+l_{W}+l_{B} \end{array}$$

### 3.5. Semi-Supervised Prediction Optimization Method

Interactive clicks represent the user’s intentions, so the segmentation results near the click locations should match the click information. However, in practical scenarios where the target region in the image is similar to the background, the target region is partially occluded, the texture is complex, or the target region is unknown, the segmentation results near the interactive clicks may not be correct and may not align with the click information. For such challenging segmentation scenarios, after model training is completed, we use user annotation data to fine-tune specific model parameters, thereby enhancing the instructive nature of user interaction information. However, this approach incurs a time cost comparable to training the entire model. Therefore, to reduce the time expense, we employ zero-order optimization.

In this paper, we formulate this problem as an optimization problem and propose a semi-supervised prediction optimization method (POM) that uses user-provided interactive clicks as supervised information to adjust the output of the GFEM module in the EnNet network model during the inference stage to ensure that the pixels in the local area near the interactive click are correctly segmented in the final segmentation result, as shown in Figure 6. In detail, let *z* be the output of the GFEM module, *g* be the processing process of the latter part of the network model, and $\hat{y}$ be the final segmentation result, which can be represented by Equation (Equation 12).
(12)$$\begin{array}{cc} \hat{y} & =g(z) \end{array}$$

This algorithm takes *z* as the optimization object. During the model inference stage, when the predicted result at the interactive point does not match the interactive information, this algorithm adjusts the *z* value by fine-tuning to change the output of the network model so that the predicted result at the interactive point matches the interactive information. The final segmentation result is corrected by adjusting the output *z* of the GFEM module.The process of correcting the final segmentation result by adjusting the output *z* of the FEM module can be formulated as an optimization problem described in Equation (Equation 13) that seeks to find the value of *z* that minimizes the loss function $L_{R}$.
(13)$$\begin{array}{cc} z^{*} & =arg\min_{z}L_{R} \end{array}$$

The optimization loss function $L_{R}$ is used to measure the error between the model’s predicted results $\hat{y}$ and the true labels *y*. In the input fusion module of the EnNet network model, the interaction points are encoded as circular disks with a radius of five pixels centered at the interaction point, and the loss function $L_{R}$ only calculates the prediction error at these circular disks to improve the segmentation accuracy in the local area near the interaction points. Let *p* be the set of interaction points; $P_{f}$ and $P_{b}$ are sets of interaction points labeled as foreground and background. For an interaction point *i* in *P* , the predicted result of the model at point *i* is denoted as $g{\left(z\right)}_{i}$; the label values at the foreground interaction point and the background interaction point are 1 and 0. The prediction loss for the foreground interaction points and the background interaction points are shown in Equations (Equation 14) and (Equation 15), respectively.
(14)$$\begin{array}{cc} l_{f} & =\sum_{i\in P_{f}}{\left[1-g{\left(z\right)}_{i}\right]}^{2} \end{array}$$
(15)$$\begin{array}{cc} l_{b} & =\sum_{i\in P_{b}}{\left[g{\left(z\right)}_{i}\right]}^{2} \end{array}$$

To prevent large variations of *z* during the algorithm execution, a second-order norm is introduced as a constraint term in the loss function, and the weight coefficient λ is fixed at 0.001. Therefore, the optimization loss function is shown in Equation (Equation 16), where $z^{0}$ represents the initial value of *z*.
(16)$$\begin{array}{cc} L_{R}\left(z\right) & =l_{f}+l_{b}+\lambda {∥z-z^{0}∥}_{2} \end{array}$$

We minimize the prediction loss function $L_{R}$ and obtain the optimal $z^{*}$ with the L-BFGS algorithm. In each iteration of the L-BFGS algorithm, the gradient of the prediction loss function $L_{R}$ with respect to *z* needs to be calculated using the back-propagation algorithm applied to the latter part of the network model. This process requires multiple derivative computations using the chain rule and can be time-consuming. To reduce the time cost, the proposed prediction optimization method can use zero-order gradient estimation [41] to approximate the gradient of the loss function $L_{R}$ with respect to *z* without sacrificing too much accuracy. The zero-order gradient estimation method imitates the first-order method and uses finite difference to estimate the gradient using two very close points of the objective function values, f(x+hv) and f(x−hv), and a small constant *h* along the direction vector *v*. The gradient of the loss function $L_{R}$ at sample point *i* can be approximated using zero-order gradient estimation as shown in Equation (Equation 17), where *h* is a small constant and $e_{i}$ is the standard vector base with only the $i_{th}$ component equal to 1.
(17)$$\begin{array}{c} \hat{\nabla }L_{R}^{i}\left(z\right)\approx \frac{{L_{R}}^{i}\left(z+he_{i}\right)-{L_{R}}^{i}\left(z-he_{i}\right)}{2h} \end{array}$$

The prediction optimization method employs L-BFGS for multi-step iterations to obtain the optimal solution. As the iterations progress, the accuracy of the required gradients also increases. Although zero-order gradient estimation methods may introduce some errors in estimating the gradients, they can still be used to approximate the gradients within an acceptable error range, thereby reducing the time cost of the prediction optimization method.

## 4. Experiments and Results

In this section, the efficacy of GFEM, ERM, and POM has been validated through dissolution experiments, exemplifying the superiority of these methods through comparisons with other approaches. Additionally, the utilized dataset, evaluation metrics, and experimental details are presented.

### 4.1. Datasets

We trained the proposed model on the COCO + LVIS dataset, which is a combination of COCO [42] and LVIS [43]. Specifically, this dataset includes a subset of images and corresponding annotations from the LVIS dataset, supplemented with annotations from the COCO dataset, resulting in a dataset with 104,000 images and 1.6 million instance-level annotations.

We evaluate the proposed method on the GrabCut [15], Berkeley [44], DAVIS [45], and SBD datasets [46]. The GrabCut dataset contains 50 images and 50 instance-level labels; the Berkeley dataset includes 200 training images and 100 testing images, of which 96 testing images have 100 instance-level labels; the DAVIS dataset, usually used for video object segmentation, includes 50 high-quality annotated videos, from which we extract 10% annotated frames as the testing set for interactive image segmentation, consisting of 345 images and 345 instance-level labels; the SBD dataset consists of 11,355 images, including 8498 training images and 2857 validation images, of which the validation set has 6671 instance-level labels, commonly used as the testing set for interactive image segmentation.

### 4.2. Evaluation Metrics

Mean intersection over union (mIoU), the standard number of clicks (NoC), and Number of Failure (NoF) are used as evaluation metrics. NoC represents the number of interactive clicks required to achieve an IoU threshold, which is typically set to 85% and 90%, denoted as NoC@85 and NoC@90, respectively. NoF refers to the number of images in the test set that cannot reach the specified IoU threshold even after a certain number of interactive clicks, which is usually set to 90%, denoted as NoF@90.

### 4.3. Implementation Details

In this paper, we employ the interactive point simulation sampling strategy proposed in RITM to generate interaction points with a maximum of 20 points, including 1–10 positive samples and 0–10 negative samples. The COCO + LVIS dataset is used as the training set, with random scaling in the range of [0.75, 1.4] and random cropping to a size of [320, 480]. The training images are augmented by horizontal flipping, vertical flipping, edge padding, brightness jittering, contrast jittering, and color jittering. The backbone network is initialized with weights pretrained on ImageNet, while the remaining network structures are initialized using the Kaiming initialization scheme. The optimizer uses Adam, with $\beta_{1}$ set to 0.9 and $\beta_{2}$ set to 0.999. A total of 230 epochs are trained with a batch size of 64, an initial learning rate of $5\times 10^{-4}$, and a learning rate decay rate of 0.1, with learning rate decay applied at the 190th and 210th epochs. All experiments are conducted on a PC with an Intel(R) Xeon(R) Gold 6230R CPU @ 2.10GHz (Intel Corporation, Santa Clara, CA, USA) and an NVIDIA GeForce RTX 3090 (NVIDIA, Santa Clara, CA, USA) running an Ubuntu 20.04.1LTS environment.

### 4.4. Ablation Study

In this paper, we conducted an ablation study on the proposed components by using the basic segmentation network (BSM) as the baseline. The effectiveness of each component was evaluated.

GFEM: As shown in Table 1, adding GFEM to the baseline model improved the results of NoC@85, NoC@90, and NoF@90, indicating that GFEM can enhance the accuracy and convergence of the model. To assess whether the GFEM module can enhance the understanding of user interaction information and image semantics, the paper presents feature heatmaps of the model outputs with and without GFEM, as shown in Figure 7. It can be observed that the outputs with GFEM more accurately delineate the contours of the target rather than providing a rough estimate. This indicates that the inclusion of GFEM improves the understanding of user interaction information and image semantics, leading to better segmentation results. The predicted masks of baseline + GFEM are shown in Figure 8. GFEM applies self-attention mechanisms to weight the correlation between different regions, allowing the model to focus more on the information related to the target region, ignore noise information unrelated to the target region, extract more accurate feature representations, and improve the effectiveness of the model.

ERM: As shown in Table 1, by adding ERM to both the Baseline and Baseline + GFEM models, improved results are achieved in terms of NoC@85, NoC@90, and NoF@90. However, it is worth noting that Baseline + GFEM already achieves a rough segmentation of the target region. The addition of ERM further enhances the results, indicating that ERM effectively utilizes fine-grained features to optimize the coarse segmentation output. This leads to improved accuracy and convergence of the model, reducing the number of required interactions. The predicted masks of baseline + ERM are shown in Figure 9. The best results were achieved with the combination of baseline + GFEM + ERM, which demonstrates the effectiveness of the EnNet architecture.

POM: The experiment investigated the relationship between the number of times zero-order gradient estimation is used and the segmentation accuracy under the premise of setting the iteration data buffer size *m* of the L-BFGS algorithm to 20, the acceptable error ϵ to $10^{-8}$, and the maximum number of iterations to 10. As shown in Figure 10, the experimental results show that using zero-order gradient estimation in the early iterations of the prediction optimization method will slightly decrease the segmentation accuracy, but when the number of uses exceeds a certain threshold, the segmentation accuracy will decrease significantly. Since the prediction optimization method needs to solve gradients in each iteration, the range of the number of times that zero-order gradient estimation is used is from 0 to 10. Based on the results on several test sets, zero-order gradient estimation is used to calculate the gradients in the first four iterations of the prediction optimization method, and back-propagation is used to calculate the gradients in the subsequent iterations.

After determining the number of times to use zero-order gradient estimation in the prediction optimization method, we applied the prediction optimization method on EnNet to validate its effectiveness by comparing its performance before and after applying the prediction optimization method. As shown in Table 2, Table 3 and Table 4, applying POM on EnNet improves the model’s NoC@85, NoC@90, and NoF@90, indicating that POM can improve the model’s accuracy and convergence.The predicted masks of EnNet+POM are shown in Figure 11.

### 4.5. Comparison with Other Works

We evaluate the proposed method on several commonly used interactive image segmentation datasets, including GrabCut, Berkeley, DAVIS, and SBD. As shown in Table 5, Table 6 and Table 7, our method exhibits superior performance in terms of accuracy and convergence compared with other methods. Specifically, on DAVIS and SBD, our method achieves the best NoC@85; on Berkeley, DAVIS, and SBD, our method outperforms other interactive image segmentation methods in terms of NoC@90; and on Berkeley and SBD, our method achieves the best NoF@90. On DAVIS, our method’s NoF@90 is second only to RITM. Figure 12 shows the average intersection over union (IoU) with the number of interactions on GrabCut, Berkeley, DAVIS, and SBD datasets for our method and other methods. Compared with other methods, our method achieves higher accuracy with the same number of interactions, and the IoU curve of our method has smaller fluctuations.

## 5. Conclusions

In this paper, we propose an interactive image segmentation architecture named EnNet, which employs a coarse-to-fine network design to fully exploit both global and local information. Additionally, an interactive information-guided optimization method is proposed to optimize the predicted mask during the inference stage, leading to better segmentation results. Experimental results demonstrate the effectiveness of our proposed method. In future work, we will focus on exploring more efficient network designs to reduce computational cost for better adaptation to practical application scenarios.

## Figures and Tables

**Figure 1 sensors-24-06361-f001:**
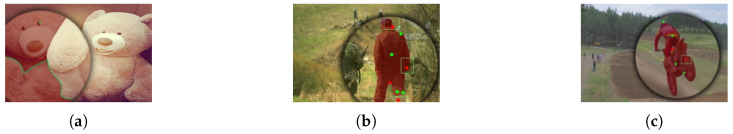
Examples of existing problems. (**a**) Over-segmentation. (**b**) Boundary blur. (**c**) Segmentation of unknown objects. The problematic area is highlighted by a green box.

**Figure 2 sensors-24-06361-f002:**
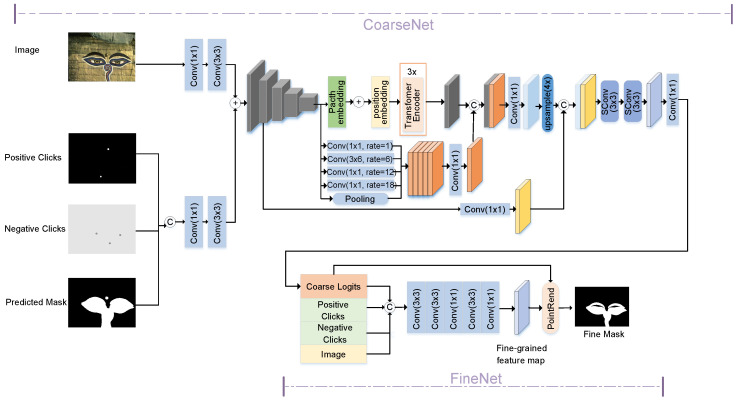
Overview of EnNet. The EnNet is composed of two parts: CoarseNet and FineNet. The CoarseNet takes the input image and interactive clicks as input to produce coarse logits. These coarse logits are then used by the FineNet to improve and refine the mask. ‘+’ represents ‘add’, and ‘C’ represents ‘concat.’

**Figure 3 sensors-24-06361-f003:**
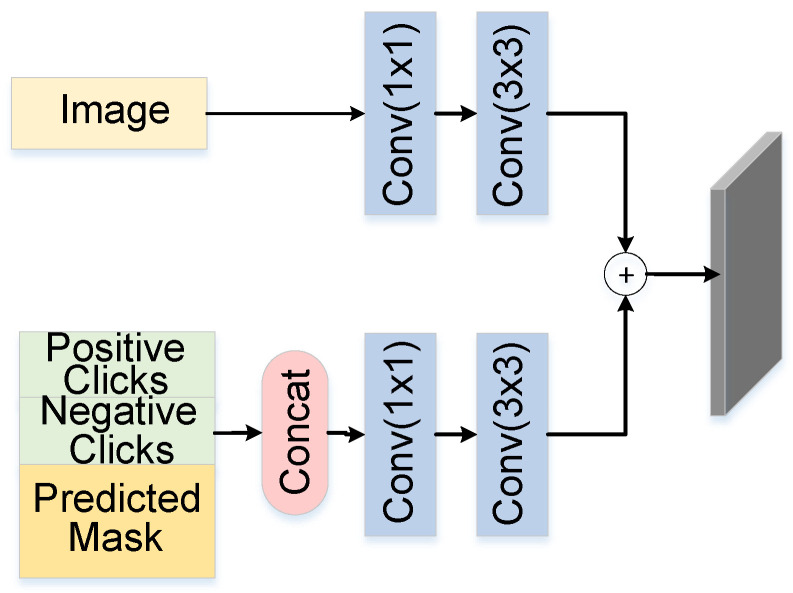
Overview of input. This figure illustrates how the image and user interactions (positive/negative clicks, predicted mask) are processed and combined using convolutions, forming the input for segmentation refinement.

**Figure 4 sensors-24-06361-f004:**
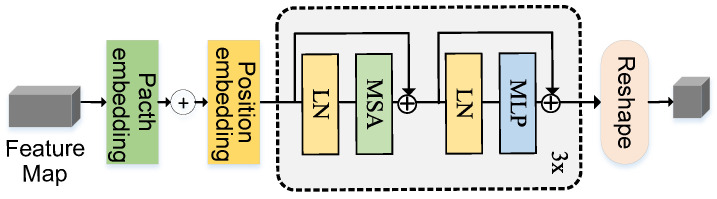
Overview of GFEM. It starts by embedding the input feature map into patches and adding positional encoding to retain spatial information. The data are then processed through three layers of multi-head self-attention (MSA) and multilayer perceptrons (MLP), each followed by layer normalization (LN). Finally, the output is reshaped back into a feature map for the next stage of processing. The GFEM is designed to enhance global contextual information for better segmentation performance.

**Figure 5 sensors-24-06361-f005:**
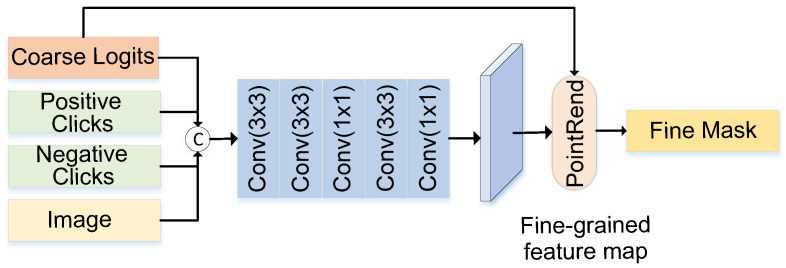
Overview of ERM. The input consists of coarse logits, positive/negative clicks, and the image itself, which are concatenated. These inputs are passed through several convolution layers to extract fine-grained features. The PointRend module then uses these features to refine the boundaries and produce a more accurate fine mask for the segmentation task. The ERM is designed to improve boundary accuracy by leveraging fine details in the feature map.

**Figure 6 sensors-24-06361-f006:**
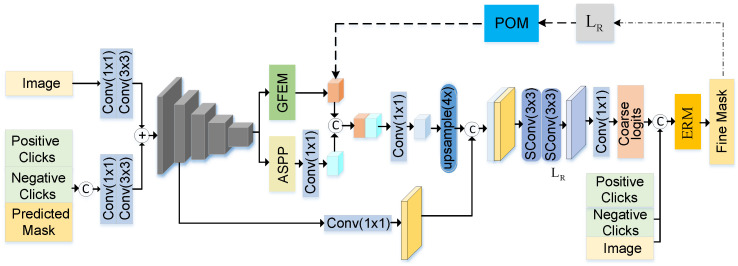
Overview of Semi-supervised prediction optimization method. The method uses user-provided interactive clicks as supervised information to adjust the output of the Global Feature Enhancement Module (GFEM) during inference. By minimizing the loss function $L_{R}$, POM ensures that pixels near the interactive clicks are correctly segmented in the final result. The process includes feature extraction, coarse logits generation, and boundary refinement through the Edge Refinement Module (ERM), ultimately improving segmentation accuracy in local regions around the clicks.

**Figure 7 sensors-24-06361-f007:**
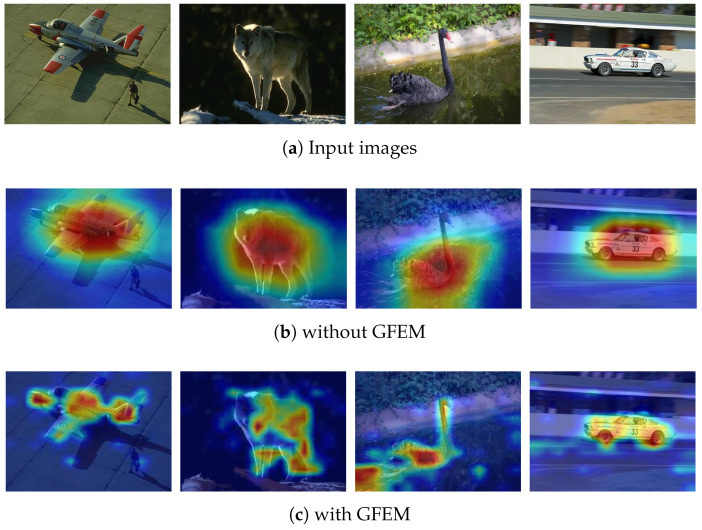
Heat maps with and without GFEM. This figure compares the heat maps of input images processed without (**b**) and with (**c**) the Global Feature Enhancement Module (GFEM). The heat maps without GFEM (**b**) show less accurate attention to the target object, while the maps with GFEM (**c**) demonstrate better focus and delineation of the object’s boundaries, improving segmentation accuracy by enhancing the global contextual information.

**Figure 8 sensors-24-06361-f008:**
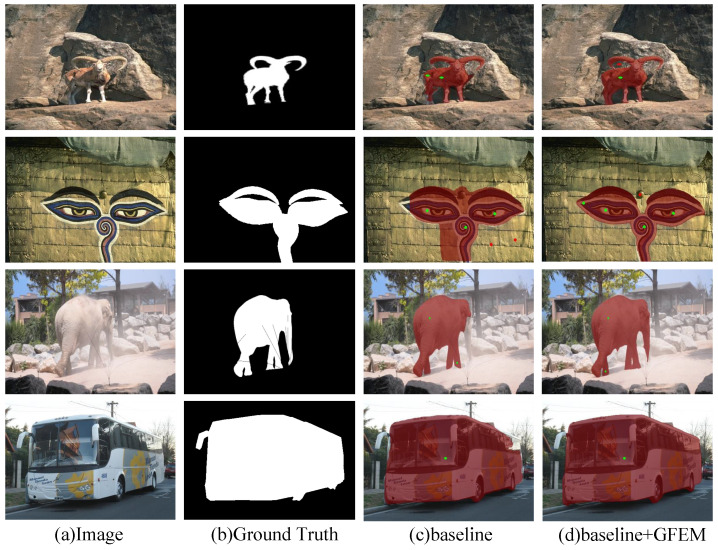
Predicted Masks with and without GFEM. This figure compares the predicted segmentation masks using the baseline model (**c**) and the baseline model with the Global Feature Enhancement Module (GFEM) (**d**). The GFEM-enhanced model (**d**) shows improved boundary delineation and overall segmentation quality compared with the baseline alone (**c**). The green dots in the figure represent positive points, while the red dots represent negative points. This color coding is maintained in all subsequent figures.

**Figure 9 sensors-24-06361-f009:**
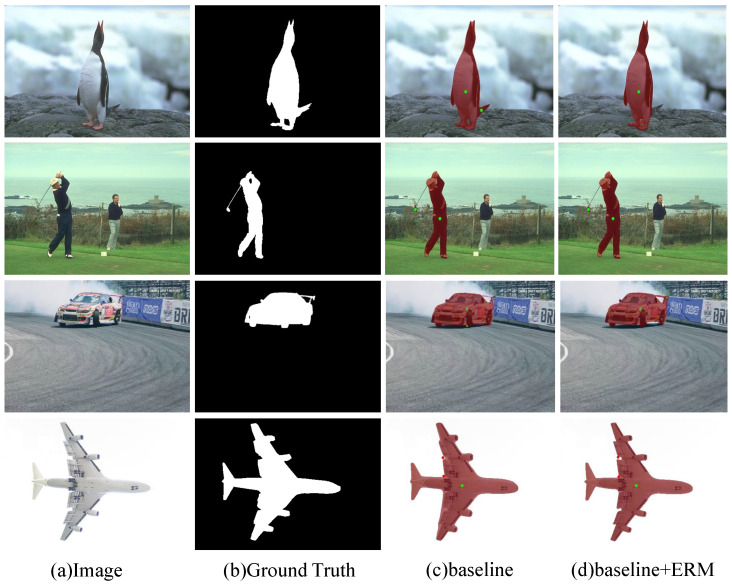
Predicted Masks with and without ERM. This figure compares the predicted segmentation masks using the baseline model (**c**) and the baseline model with the Edge Refinement Module (ERM) (**d**). The ERM-enhanced model (**d**) produces more precise and detailed segmentation results compared with the baseline (**c**), especially in challenging areas like fine edges.

**Figure 10 sensors-24-06361-f010:**
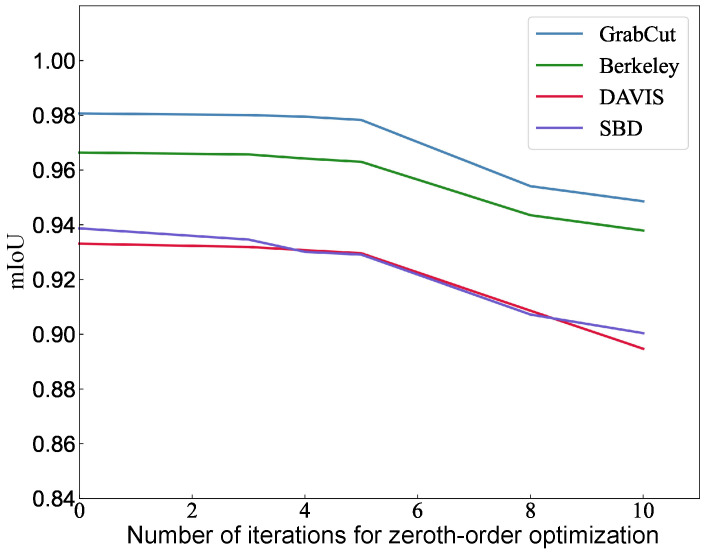
mIoU under different zero-step estimation usage times. This figure illustrates the change in mean Intersection over Union (mIoU) across various datasets (GrabCut, Berkeley, DAVIS, SBD) as the number of iterations for zero-order optimization increases. The plot shows that mIoU generally decreases as the number of iterations increases, indicating a potential loss in segmentation accuracy with prolonged zero-order optimization.

**Figure 11 sensors-24-06361-f011:**
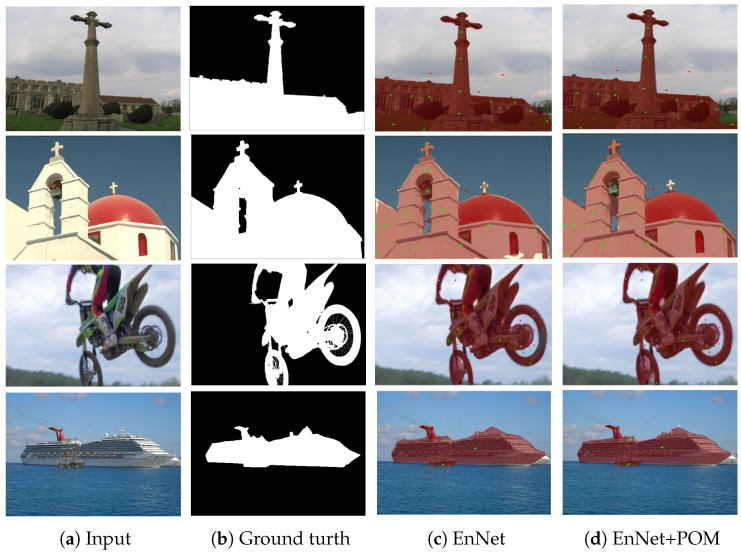
Predicted mask with or without prediction optimization method. This figure compares predicted masks from EnNet (**c**) and EnNet with POM (**d**) against the ground truth (**b**). The addition of POM improves segmentation accuracy, especially in complex areas, leading to results closer to the ground truth.

**Figure 12 sensors-24-06361-f012:**
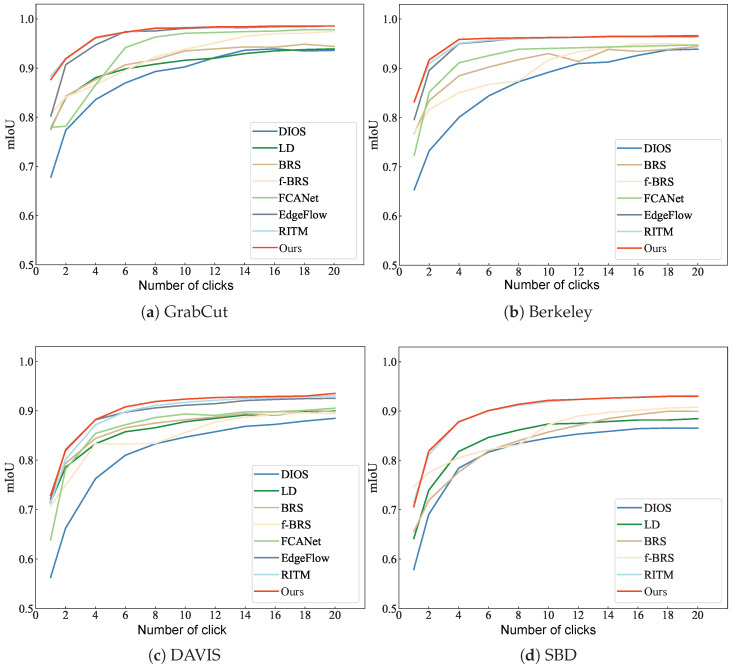
MIoU changes with number of clicks on GrabCut, Berkeley, DAVIS and SBD.

**Table 1 sensors-24-06361-t001:** NoC@85, NoC@90, NoF@90 under different modules. This table shows the performance of different module combinations on various datasets. Lower NoC@85, NoC@90 values indicate fewer clicks needed to reach 85% and 90% IoU, while lower NoF@90 means fewer failed cases. The GFEM+ERM combination performs best, requiring fewer clicks and having the lowest failure rates.

	NoC@85			NoC@90				NoF@90		
	**GrabCut**	**DAVIS**	**SBD**	**GrabCut**	**Berkeley**	**DAVIS**	**SBD**	**Berkeley**	**DAVIS**	**SBD**
Baseline	1.67	4.77	4.07	1.64	2.30	5.58	6.09	3	65	791
Baseline + GFEM	1.52	4.36	3.81	1.64	2.30	5.58	6.09	0	61	756
Baseline + ERM	1.60	4.50	3.88	1.68	2.39	5.67	6.18	0	63	770
Baseline + GFEM + ERM	1.49	4.24	3.72	1.59	2.20	5.45	5.87	0	59	739

**Table 2 sensors-24-06361-t002:** NoC@85 with or without prediction optimization method.

	GrabCut	DAVIS	SBD
EnNet	1.49	4.24	3.72
EnNet + POM	1.46	4.07	3.60

**Table 3 sensors-24-06361-t003:** NoC@90 with or without prediction optimization method.

	GrabCut	Berkeley	DAVIS	SBD
EnNet	1.59	2.20	5.45	5.87
EnNet + POM	1.55	2.08	5.26	5.67

**Table 4 sensors-24-06361-t004:** NoF@90 with or without prediction optimization method.

	Berkeley	DAVIS	SBD
EnNet	0	59	739
EnNet + POM	0	57	722

**Table 5 sensors-24-06361-t005:** NoC@85 of different methods.

Method	GrabCut	DAVIS	SBD
DIOS	5.08	9.03	9.22
LD	3.20	5.05	7.41
BRS	2.60	5.58	6.59
f-BRS	2.50	5.39	5.06
FCANet	2.18	5.54	-
EdgeFlow	1.60	4.54	-
CDNet	2.22	5.17	4.37
RITM	1.42	4.36	3.80
Ours	1.46	4.07	3.60

**Table 6 sensors-24-06361-t006:** NoC@90 of different methods.

Method	GrabCut	Berkeley	DAVIS	SBD
DIOS	6.04	8.65	12.58	12.80
LD	4.79	-	9.57	10.78
BRS	3.60	5.08	8.24	9.78
f-BRS	2.98	4.34	7.81	8.08
FCANet	2.62	4.66	8.83	-
EdgeFlow	1.72	2.40	5.77	-
CDNet	2.64	3.69	6.66	7.87
RITM	1.54	2.60	5.74	6.06
Ours	1.55	2.08	5.26	5.67

**Table 7 sensors-24-06361-t007:** NoF@90 of different methods.

Method	Berkeley	DAVIS	SBD
BRS	10	77	-
FCANet	-	87	-
f-BRS	2	78	1466
CDNet	-	65	-
RITM	2	52	811
Ours	0	57	722

## Data Availability

The original contributions presented in the study are included in the article; further inquiries can be directed to the corresponding author.

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
