# Peer review of "EnNet: Enhanced Interactive Information Network with Zero-Order Optimization"

_sensors, 2024, doi:10.3390/s24196361_

Round 1

Reviewer 1 Report

Comments and Suggestions for Authors

When I read the paper, the summary is not clear to me because I cannot understand it, and when I read it in depth I think that tests should be done with medical images, since all medical images are grayscale, and on the contrary, the images shown in the paper are color images that certainly improve the possibility of segmentation. I have performed segmentation of images with tumors and I have checked some current papers and it is not easy, so in my opinion it would be more interesting to show this in grayscale images than in color images. The methodology seems clear but since there are no clear examples of this technique on medical images, I would say that it is better.

Author Response

Dear Reviewer, Thank you for your valuable feedback and thoughtful suggestions. We appreciate your insights and would like to address your points as follows:   Our work aims to segment the objects users want to extract from natural images more quickly and accurately, helping users who need to create custom datasets or perform tasks like image cutouts. In most natural images, the boundaries between different objects are quite distinct. Therefore, our model focuses on understanding user interaction information, which enables the segmentation of objects that the model has not been specifically trained on. However, in medical images, the segmentation targets—such as lesions and organs—are relatively fixed and specific, so network designs for medical images often prioritize object recognition over user interaction. This differs from the main goal of our work.   Best regards.

Reviewer 2 Report

Comments and Suggestions for Authors

The paper introduces EnNet, a novel architecture for interactive image segmentation that integrates user interaction information more effectively by using attention mechanisms and a coarse-to-fine approach. The method also employs a semi-supervised prediction optimization algorithm with zero-order optimization, aiming to reduce computational overhead while maintaining segmentation accuracy. The authors validate their approach through extensive experiments on various datasets, demonstrating improvements in terms of accuracy and efficiency.

Some suggestions are as follows:

1.  The proposed network architecture and optimization method are clearly articulated.  However, the paper would be strengthened by a more in-depth discussion on the rationale behind the coarse-to-fine strategy and its specific advantages over previous methods.

2.  Offer a deeper analysis of the trade-offs between zero-order optimization and traditional methods, and discuss EnNet’s performance under varying interaction complexities.

Expand Discussion: Include a more in-depth analysis of the trade-offs between zero-order optimization and traditional methods, and discuss EnNet's performance under varying levels of interaction complexity.

Comments on the Quality of English Language

it is ok

Author Response

Dear Reviewer, Thank you for your valuable feedback and thoughtful suggestions. We appreciate your insights and would like to address your points as follows:   1. Coarse-to-Fine Strategy: We agree that a more detailed discussion on the rationale behind the coarse-to-fine approach would strengthen the paper. The coarse-to-fine strategy was chosen because it allows the network to first capture a broader, less detailed understanding of the image, which reduces over-segmentation errors in complex scenes. Fine-tuning then helps refine the boundaries of the segmented object, resulting in a more precise mask, particularly in areas where fine-grained details are critical. Compared to previous methods, this approach balances speed and accuracy by avoiding the need for highly detailed computations at every stage, focusing computational resources on where they are most needed.   2. Trade-offs with Zero-Order Optimization: We will expand the discussion on the trade-offs between zero-order optimization and traditional optimization methods. Zero-order optimization was selected to reduce the computational burden without requiring gradients for each parameter, making it faster for certain types of segmentation tasks.   We will revise the manuscript to include a deeper analysis in these areas, ensuring that the advantages of our approach and its performance in different interaction settings are more clearly articulated.   Best regards.

Reviewer 3 Report

Comments and Suggestions for Authors

This paper designs an EnNet to pay attention to interactive information mining from the network architecture and optimization procedure.  For the optimization problem, this paper proposes uses  zeroth-order optimization during the first four iterations of training.  This can reduce computations significantly. The paper is well organized. Here are some comments to improve the quality of this paper.

1、The motivation should be highlighted in the Section 1.

2、In the last paragraph, there is no Section II, Section III, they are should be Section 2, Section 3.

All the number should be checked carefully to avoid the typos.

3、Some comments should be highlighted in Section 2.  It is suggested to not use lengthy paragraph.

4、It seems that the titles of figures are simple. It is suggested to give detailed descriptions. 

Comments on the Quality of English Language

The English should be improved. 

Author Response

Dear Reviewer,   Thank you for your positive feedback and detailed suggestions. We appreciate your time and effort in reviewing our work, and we would like to address your comments as follows:  
  1. Highlighting the Motivation in Section 1: We agree that the motivation for our work should be more prominently highlighted in the introduction. We will revise Section 1 to more clearly emphasize the specific challenges in interactive image segmentation that our method addresses, such as the computational burden of traditional methods and the need for effective utilization of user interaction information. This will ensure that the readers understand the core reasons driving the development of EnNet.

  1. Section Numbering and Typos: We appreciate your attention to detail regarding the section numbering. We will carefully review and correct the numbering of sections.

  1. Comments in Section 2: We will revise Section 2 to break up lengthy paragraphs and make the discussion more concise and focused. This will improve the readability and clarity of the content, ensuring that key points stand out more effectively.

  1. Figure Descriptions: We will revise the figure captions to provide more detailed and informative descriptions, ensuring that each figure clearly illustrates the point it is meant to convey. This will help readers better understand the relevance and importance of each figure in relation to the overall work.
  We appreciate your constructive feedback and will incorporate these changes to improve the quality and clarity of the paper.   Best regards.

Round 2

Reviewer 1 Report

Comments and Suggestions for Authors

Thanks is ok